# Effect of Varying Steel Fiber Content on Strength and Permeability Characteristics of High Strength Concrete with Micro Silica

**DOI:** 10.3390/ma13245739

**Published:** 2020-12-16

**Authors:** Babar Ali, Rawaz Kurda, Bengin Herki, Rayed Alyousef, Rasheed Mustafa, Ahmed Mohammed, Ali Raza, Hawreen Ahmed, Muhammad Fayyaz Ul-Haq

**Affiliations:** 1Department of Civil Engineering, COMSATS University Islamabad—Sahiwal Campus, Sahiwal 57000, Pakistan; babar.ali@cuisahiwal.edu.pk (B.A.); m.fayyaz@cuisahiwal.edu.pk (M.F.U.-H.); 2Department of Highway and Bridge Engineering, Technical Engineering College, Erbil Polytechnic University, Erbil 44001, Iraq; Hawreen.a@gmail.com; 3Scientific Research and Development Center, Nawroz University, Duhok 42001, Iraq; 4CERIS, Civil Engineering, Architecture and Georresources Department, Instituto Superior Técnico, Universidade de Lisboa, Av. Rovisco Pais, 1049-001 Lisbon, Portugal; 5Department of Civil Engineering, College of Science and Engineering, Bayan University, Erbil 44001, Iraq; bengin.awdel@bnu.edu.iq; 6Department of Civil Engineering, Faculty of Engineering, Soran University, Soran 44008, Iraq; 7Department of Civil Engineering, College of Engineering, Prince Sattam bin Abdulaziz University, Alkharj 16273, Saudi Arabia; r.alyousef@psau.edu.sa; 8Department of Environmental Engineering, College of Engineering, Knowledge University, Erbil 44001, Iraq; rasheed1954@yahoo.com; 9Civil Engineering Department, College of Engineering, University of Sulaimani, Sulaymaniyah 46001, Iraq; ahmed.mohammed@univsul.edu.iq; 10Department of Civil Engineering, Pakistan Institute of Engineering and Technology, Multan 66000, Pakistan; aliraza@piet.edu.pk

**Keywords:** mechanical properties, fiber-reinforced concrete, permeability, durability, tensile strength, micro-silica/silica fume, steel fiber

## Abstract

For the efficient and durable design of concrete, the role of fiber-reinforcements with mineral admixtures needs to be properly investigated considering various factors such as contents of fibers and potential supplementary cementitious material. Interactive effects of fibers and mineral admixtures are also needed to be appropriately studied. In this paper, properties of concrete were investigated with individual and combined incorporation of steel fiber (SF) and micro-silica (MS). SF was used at six different levels i.e., low fiber volume (0.05% and 0.1%), medium fiber volume (0.25% and 0.5%) and high fiber volume (1% and 2%). Each volume fraction of SF was investigated with 0%, 5% and 10% MS as by volume of binder. All concrete mixtures were assessed based on the results of important mechanical and permeability tests. The results revealed that varying fiber dosage showed mixed effects on the compressive (compressive strength and elastic modulus) and permeability (water absorption and chloride ion penetration) properties of concrete. Generally, low to medium volume fractions of fibers were useful in advancing the compressive strength and elastic modulus of concrete, whereas high fiber fractions showed detrimental effects on compressive strength and permeability resistance. The addition of MS with SF is not only beneficial to boost the strength properties, but it also improves the interaction between fibers and binder matrix. MS minimizes the negative effects of high fiber doses on the properties of concrete.

## 1. Introduction

Plain cement concrete (PCC) is the most versatile construction material owing to its multiple benefits i.e., high compressive strength, cost-efficient, in-situ formability, thermal and electrical insulation, imperviousness, etc. Ingredients and mix design of PCC can be changed to obtain different types of concrete that suit different structural loadings and environments. To further raise the importance of conventional concrete, several performance-related issues need to be resolved. PCC generally performs weakly under tensile loadings. Its splitting tensile strength (STS) is very low compared to its compressive strength (CS). According to Ali and Qureshi [1,2] and Koushkbaghi et al. [3] PCC has a STS/CS ratio of about 7–9.5%. Zain et al. [4] showed that the STS/CS ratio decreases as the strength class of concrete is upgraded, therefore, high strength PCCs are more vulnerable to brittle failure than normal strength PCCs. PCC has low energy absorption capacity (or toughness) under both tensile and compressive loadings [5,6]. It undergoes a sudden failure after carrying the load beyond its peak capacity and it has very low residual strength (almost negligible compared to fiber-reinforced concrete) [7]. Mechanical performance of PCC undergoes significant degradation with time when subjected to environmental stresses (freeze-thaw) [8] and weathering actions of water and acid attack [9]. Due to the inherent weakness of PCC under tensile loadings, large structural dimensions cannot be avoided unless it is reinforced with some high strength material i.e., steel rebars, glass fiber-reinforced polymers (GFRP) bars, carbon fiber-reinforced polymer (CFRP) bars, etc. 

Nowadays, fibers are being used as a discrete 3-dimensional reinforcement to overcome the deficiency of PCC in tensile strength. With the addition of proper fiber-type in concrete, initiation and proliferation cracks under both tensile and compressive loads can be controlled or delayed. Many types of reinforcements are available commercially that own their application-specific characteristics e.g., carbon fiber, steel fiber (SF), glass fiber, polypropylene fiber [10] organic fibers [11], carbon nano-tubes [12,13] etc. Tensile reinforcements disperse throughout the PCC matrix, bridge the microcracking [12,13]. Considering the mechanical performance of concrete, SF is by far more superior fiber compared to other industrial fibers [10]. SF has a very high tensile strength of over 1200 MPa and an elastic modulus of about 200 GPa. The literature confirms its vitality as a superior reinforcement material that ensures satisfying tensile, compressive, flexural and shear strength properties [14,15,16,17]. By improving the strength per unit quantity of material, SF-reinforced concrete (SFRC) shows lower economic and environmental impact compared to PCC [14].

There are several issues relevant to the underutilization of SF in fiber-reinforced concrete that must be addressed. According to Lee et al. [18], the primary reason for the failure of SF-Reinforced Concrete (SFRC) is the failure linked to the interface between fiber and binder matrix of the concrete. Two different types of failures are linked to SFRCs when the underutilized fiber is separated from the binder matrix [19]. The first type of failure occurs at the interface between the fiber and binder matrix, whereas another type of failure occurs in the adhering binder matrix. Both of these failures lead to under-utilization of matrix and full tensile strength potential of fibers, and consequently leading to cracking of concrete. Therefore, bond strength between fibers and binder matrix plays a significant role in defining the tensile capacity of fibrous composites.

The effect of SF on the properties of the fibrous composite also depends on the dosage of fiber. There is a consensus among researchers that the tensile and bending capacity of concrete improves the increasing fiber dosage (0 to 2% by volume fraction) [3,9,20,21,22,23], but the literature has shown the mixed effects of varying SF dosage on the compressive behavior of concrete. Some studies [23,24] report that SF induces porosity into concrete therefore, compressive strength and elastic modulus of concrete undergoes degradation with the rising fiber volume. Whereas, there are studies [3,9,21] which have shown positive effects of SF on compressive behavior of concrete. The improvements in compressive strength were attributed to the increased stiffness and confinement of concrete [3,9,21]. However, a consensus is found among findings of the researchers that SF is beneficial to compression toughness of concrete [5,23,25,26]. 

There are some issues related to application of SFRC that are vital to be understood and resolved. The reasons behind the mixed effects (positive, negative or inconsiderable) of SF dose on compressive strength of concrete are still needed to be understood for the effective use of fibers under compression loadings. Moreover, poor bond strength between fiber and binder matrix reduces the utilization of full potential of SF. It is essential to investigate the influence of such materials (i.e., MS) on SFRC that help in strengthening binder matrix and improve the dispersion of fibers [2,10]. This study is designed to evaluate the effects of varying SF dosage on the properties of a high strength concrete. To investigate the role of bond strength at interfacial zone between fiber and matrix, the effect of SF dosage was also explored with and without micro-silica (MS). SF was used at six different levels i.e., low fiber volume (Vf = 0.05% and 0.1%), medium fiber volume (Vf = 0.25% and 0.5%) and high fiber volume (Vf = 1% and 2%). Each dose of SF was investigated with 0%, 5% and 10% MS as by volume of binder. MS is an excellent consumer of portlandite CH (in pozzolanic reaction, that strengthens the binder matrix and improves the bond strength of fibers [27]. Concrete mixtures were assessed based on the results of basic mechanical and permeability tests. The results of this study provide a useful information on the selection of SF dose for the optimum mechanical and durability performance. Moreover, experimental results favor the use of MS to maximize the utilization of SF under compressive and tensile forces.

## 2. Materials and Methods 

### 2.1. Materials

Conventional and supplementary materials used for the production of concrete mixtures are explained in this section. Type I general purpose cement (Bestway 53 Grade, Haripur, Pakistan) was used as the main binder as per specifications of ASTM C150 [28]. Micro-silica (also known as silica fume obtained from Jaza Minerals, Karachi, Pakistan) containing 90–94% silica was used as a partial cement substitute. Properties of cement and micro-silica can be assessed from the study of Ali et al. [2]. Siliceous sand from the Lawrancepur quarry (Attock, Pakistan) was used as fine aggregate. Dolomite sandstone of Kirana-hills of Sargodha was used as the coarse aggregate. The maximum aggregate size of coarse and fine aggregate is 12.5 mm and 4.75 mm. The distribution of different particle sizes in both coarse and fine aggregates is shown in Figure 1. Characteristics of aggregates are given in Table 1.

Hooked steel fiber (SF) was studied as a discrete reinforcement in concrete. It has tensile strength of 750 MPa and a density of 7750 kg/m^3^. It was sourced from Dramix© (Zwevegem, Belgium). SF has a length of 30 mm and a diameter of 0.38 mm. Its overview is shown in Figure 2. To control the loss of workability with a rising dose of SF, Viscorete 3110 (Sika Chemicals, Lahore, Pakistan) was used as an ultra-high range plasticizer. Tap water free from organic/inorganic impurities was used during the mixing and for curing as well.

### 2.2. Design of Concrete Mixtures

A total of 21 concrete mixes were studied in this research. Six different doses (Vf = 0.05, 0.1, 0.25, 0.5, 1 and 2%) of SF were used to produce fiber-reinforced concretes. SF volumes were selected to observe the effects of a wide range of fiber doses on the properties of concrete. Fibers were used as by volume fraction of concrete (i.e., 1% Vf of SF = 78 kg of SF). Each fiber dosage is investigated with 0%, 5% and 10% micro-silica (MS). MS serves as the matrix-strengthening agent by producing calcium silicate hydrate gels from the free portlandite-CH a by-product from the hydration of cement. Plain concretes were also produced with and without MS and served as reference mixes. Composition and mix proportioning of all concrete mixtures are provided in Table 2. 

Mixing of all concretes was done in a mechanical mixer having adjustable rotation speed. In the first stage, aggregates and binder were dry mixed for 3 min at speed of 40 rpm. In the second stage, half of the mixer and water reducer were added to the mix and blending continued for 3 min at speed of 40 rpm. In the third stage, the remaining water was added to the mix, and mixing was done a high speed of 80 rpm for 2 min. In the last stage, SF was added to the concrete mix, and blending continued for the next 4 min at 80 rpm. After mixing, a slump test was performed to check the desired workability of mixes (i.e., slump between 80 and 110 mm). Mixer continued to run at a slower speed of 20 rpm until the casting of specimens was completed.

### 2.3. Sample Preparation and Testing Techniques

All specimens were cast in standard steel molds and protected with a waterproof membrane for 24 h setting immediately after casting. After setting, specimens were cured in tap water for 28-days at room temperature conditions. All mixes were tested for three important mechanical parameters i.e., compressive strength (CS), modulus of elasticity (MOE), and splitting tensile strength (STS). For CS, 100 φ mm × 200 mm cylindrical specimens were tested as per ASTM C39 [29]. To determine CS, specimens were tested under compressive-hydraulic press at the rate of 0.3 MPa/s. The static MOE of each mix was determined according to ASTM C469 [30]. MOE test was conducted on the specimens of 150 φ mm × 300 mm at the stress-rate of 0.15 MPa/s. The strain data (deformation characteristics) was recorded using compressometer-extensometer. To evaluate STS, 100 φ mm × 200 mm specimens were tested following ASTM C496 [31]. The splitting-load was applied at the stress rate of 0.015 MPa/s on the specimen to determine STS. All mechanical tests were performed in a controls compression testing machine with a loading capacity of 3000 kN. To understand the effects of varying SF and MS contents on the durability of concrete, two permeability-related durability indicators were evaluated i.e., water absorption and chloride ion penetration. To test for water absorption (WA) capacity, 100 φ mm × 50 mm concrete disc specimen of each mix was tested following ASTM C642 [32]. To determine chloride ion penetration (CIP) resistance of each mix, an immersion technique was adopted as explained by the authors [2]. For the CIP test, a 100 mm × 100 mm cylindrical specimen was first cured in normal water for 28 days. Then the specimen was immersed in a 10% NaCl solution for 56 days. After conditioning in chloride solution, the specimen was split, and the failed surface of the specimen was sprayed with 0.1 N AgNO_3_ solution to observe the depth of CIP. The further detailed procedure for CIP testing can be assessed from studies [2,33]. All the results presented in this research are the mean values of the three results of each concrete mixture. The schedule of casting and testing is shown in Table 3.

## 3. Results and Discussion

### 3.1. Compressive Strength (CS)

Figure 3 shows the effect of varying SF dose on the CS of concrete. Figure 3b shows the net age change in CS with the varying SF dose. These results show a mixed effect of SF on CS at different doses. CS goes on increasing when the SF dose changed from 0 to 0.25%. Further increasing SF beyond 0.25%, CS starts reducing, and at 2% SF, CS of fibrous concrete is lesser than that of the plain concrete. Three different causes contribute to CS property due to the inclusion of fibers. The first cause is related to the confinement effect of fibers that increases the stiffness of concrete and it is known to positively affect the CS [10,34,35]. The second phenomenon is related to the entrainment of additional ITZs in concrete that has a detrimental effect on the CS. The introduction of a high number of ITZs contributes to porosity and permeable channels into the concrete and ITZs act as a weak link in the fibrous composite. The third phenomenon pertains to the resistance of cracking to the propagation of micro and macro-cracks; thus, it is known to improve the compressive stiffness of concrete. The first and third phenomenon prevails at 0.1–0.25% dose of SF, therefore, CS shows improvement due to fiber addition, whereas, at high fiber doses, a high number of ITZs introduction facilitate crack propagation and it adds to the total porosity of concrete.

Figure 4 shows the effect of MS on CS results of high strength concrete at different doses of SF. Figure 4b shows the effect of varying SF dose on CS at the levels of 0%, 5% and 10% MS. MS shows a positive effect on compressive strength concrete due to its ability to produce calcium silicate hydrate gels in pozzolanic reactive with free portlandite. The strengthening of the binder leads to improvement in the bond strength of fibers and matrix, that is why a clear difference (Figure 4b) between “net change” of SF mixes with and without MS. For example, at 0.25% SF the net changes in CS at 0%, 5% and 10% MS are 5.4%, 7.03% and 10.37%, respectively. MS also minimizes the negative effect of high fiber volumes (i.e., 2% SF) on CS. This can be credited to the strengthening of the bond at ITZ, which enhances the utilization of fibers in compression. It is confirmed from the results that the combined incorporation of 10%MS and 0.5%SF can increase the CS by more than 20%. It is verified by the literature that MS addition does not only contribute to the bond strength of fibers but it also improves the dispersion of fibers [27,36,37]. Therefore, it can be said that SF and MS have synergistic effects on the properties of concrete.

### 3.2. Modulus of Elasticity (MOE)

Figure 5 shows the effect of SF on the MOE of concrete. MOE linearly increases when the SF dose changes from 0 to 0.5%. Figure 5b shows the effect of SF dose on the net change in MOE. The improvements in MOE at 0.05–0.5% can be ascribed to increment in the confinement of specimen under compression that helps in the utilization of the full potential of the concrete matrix. Beyond 0.5%SF, MOE starts degrading similar to CS. This shows that for given high-strength concrete the optimum dose of SF for optimum MOE is 0.5%. As already explained, high fiber doses can increase the number of ITZs in concrete which leads to the reduction in compression stiffness of concrete. A slight increase in porosity of concrete due to fibers (higher than 0.5%) can also damage the MOE considerably [38]. This finding is in line with the study of Xie et al. [24]. It was observed that during compression testing, mixes with high fiber doses showed more ductile failure before collapsing completely unlike the mixes with smaller doses. A linear increase in energy absorption capacity was observed with the rise in SF dose. Ou et al. [39] reported that the main role of SF is prominent in compression toughness of concrete (post-peak load behavior) because, before peak load in the determination of MOE, fibers are not activated.

In Figure 6, the effect of MS content is shown on the MOE of concrete. Figure 6b shows the net change in MOE due to varying dose of SF at 0, 5 and 10% replacement levels of MS. A clear improvement is noticed in the MOE of concrete due to MS addition. This is credited to (1) the improved packing density of binder particles and (2) the pozzolanic reaction that consumes free lime. MOE concrete with 10% MS is 11% higher than that of the plain concrete without MS. Figure 6b shows that MS enhances the utilization of fibers. Moreover, MS minimizes the negative effect of high SF volume on MOE. The combined incorporation of MS and SF shows synergistic behavior. For example, 0.5%SF and 10% MS individually leads to improvement of 3.4% and 7%, respectively. But simultaneous incorporation of both MS and SF improves the MOE by 14.8%. This is true for all mixes made with both MS and SF.

From the results of CS and MOE, it is quite clear that both of these mechanical properties show a similar response to varying SF and MS contents. Therefore, both parameters can predict each with great accuracy. Since MOE is difficult to determine in the laboratory; therefore, it is predicted usually from CS. The relationship between MOE and half power of CS is shown in Figure 7. This relationship (Equation (1)) is drawn without considering the impact of SF or MS content:(1)$$\mathrm{MOE}=7007\sqrt{\mathrm{CS}}-18500$$
where MOE = modulus of elasticity (MPa); CS = compressive strength (MPa).

### 3.3. Splitting Tensile Strength (STS)

STS in an estimate of true tensile strength of concrete. Due to the complexity of measuring the true tensile strength under the direct tension test, STS provides a simpler measurement of the tensile strength of cementitious materials. Figure 8 shows the effect of varying SF content on STS. Unlike results of CS and MOE, STS does not show a mixed response to increasing the dose of SF. This is because, under tensile load, fibers become active way before the failure at peak load; therefore, stretching action on concrete is resisted by both concrete matrix and fibers. Figure 8 shows that the net change in STS due to SF addition is very huge compared to that observed in the results of MOE and CS. STS achieves more than 3 times positive gain compared to CS and MOE at each dose of SF. This confirms that fibers are more useful in tensile stiffness than they are in the compressive stiffness of concrete.

MS addition provides a little advancement in the tensile strength, see Figure 9. Since MS strengthens the binder matrix, some small improvements can be anticipated in the STS. The filling effect of MS particles cannot contribute to STS, only the pozzolanic reaction between portlandite and silica strengthens the concrete matrix against tensile stresses [40]. A clear view of the synergistic effect of MS and SF on the STS can be seen in Figure 9. MS addition improves the net gain due to fibers by more than 30%. Densification of the matrix leads to an efficient transfer of tensile stresses to fiber-filaments; thus, MS addition improves the utilization of fibers. The results show that using MS along with SF can help in yielding 20% more STS than that could be achieved without MS. These results have important implications for fiber-reinforced concrete/composites. Since fibers are very expensive materials, their full utilization is very necessary to design cost and performance efficient structures. Therefore, MS and other high-performance mineral admixtures can help in enhancing the utilization of fibers.

For plain concrete, STS can be fairly correlated with CS or MOE. But for fibrous concrete STS cannot be correlated with CS or MOE, see Figure 10. As, activation of fibers during compression mostly starts near or after the peak load loads; therefore, fibers do not contribute a great deal towards the advancement of CS or MOE. Whereas, under tension, fibers activate way before peak load; therefore, concretes show a huge STS change with fiber addition. Under tension, fibers do not only contribute to the peak strength of concrete, but they are also useful in the post-peak load resistance. CS, MOE, and STS of each mix are correlated in Figure 10, without considering the role of SF dose. This surface plot shows a general trend that each mechanical parameter is directly proportional to each other but with a huge scatter (R2 < 0.6).

### 3.4. Water Absorption (WA)

WA capacity of concrete represents its water-permeable volume of voids. High WA generally indicates high porosity. The effect of SF on the WA capacity of each mix is shown in Figure 11. WA undergoes mixed changes with the rising dose of SF. Small fractions of SF cause minor reductions in the WA capacity of concrete, whereas, at high doses, SF, WA absorption of fibrous concrete is slightly higher than that of the plain concrete. Both positive [21,41] and negative [42] effects of SF on WA has been reported in the literature. No study in the literature has examined the permeability characteristics of SF-reinforced concretes considering a wide range of fiber dosage. Fibers can control micro-cracking during the evolution of cementitious compounds in concrete. These can restrict the plastic and temperature shrinkage cracking which ultimately improves the permeability resistance of concrete. At the same time, fiber addition increases the number of ITZs in concrete. Poor bond at ITZs favor permeability, hence it increases of WA capacity. Apparently, at low fiber volumes, controlled shrinkage leads to reduction in WA capacity and the role of ITZs is not very dominant at low fiber volumes. But as the fiber volume increases, the number of weak ITZs favor permeability and increase the WA. The minimum WA is observed at 0.1% SF, whereas maximum WA is noticed at 2% SF.

Figure 12 shows the effect of MS on WA capacity of concrete. MS brings down the WA capacity of concrete significantly. As extremely fine particles of MS fill the gaps left between cement particles, the overall density of matrix undergoes improvement. MS can reduce the pore-size at the ITZ between fiber and matrix. MS can nullify the negative effect of SF on WA. These results implicate an important role of MS in fibrous concretes. Since, fibers at medium to high volumes (0.5–2%), increase the permeability which may favor the corrosion of SF. Corrosion of SF will significantly lower the performance of fibrous concrete over time. Therefore, the conjunctive use of fibers and MS can increase the durability life of fibrous concrete composites.

### 3.5. Chloride Ion Penetration (CIP)

Figure 13 shows the effect of SF content on the CIP of concrete. CIP results also experience changes similar to WA with the variation of SF content. CIP undergoes reduction when fiber dose changes from 0 to 0.1%. Since there is no involvement of forced electrical transfer of chloride ions in the immersion technique high conductivity of SF does not play any role in determining the CIP resistance of concrete. CIP resistance improvement at low fiber volumes can be ascribed to a reduction in the WA capacity of concrete. On the other hand, reduction in CIP resistance at high fiber volumes (1% and 2%) can be blamed to an increase in porosity or absorption capacity of the matrix. At 2%SF, CIP of concrete is about 18% higher than that of the plain concrete. Since chloride-induced corrosion is usually experienced in most concrete structures, low chloride permeability resistance of fibrous concretes (especially with a high volume of fibers) can create durability issues which must be considered while designing a concrete mix.

Figure 14 shows the effect of MS content on the CIP. The addition of 5%MS and 10%MS brings down the CIP by 23% and 33%, respectively w.r.t plain concrete (without MS). The behavior of WA and CIP with the addition of MS is very similar because MS substantially reduces the volume of permeable voids [2]. As fibrous concretes struggle with the issue of low CIP resistance at high fiber doses, MS can be a befitting addition to enhance the imperviousness of concrete. With 5 or 10% MS, high fiber volume concretes show lower CIP than control concrete (see Figure 14a).

By constricting the microchannels across the ITZs at fibers, MS can efficiently minimize the degrading effect of fibers on CIP. Almost all engineering properties of concrete depend on the growth and density of microstructure i.e., strength and permeability characteristics. CS, CIP, and WA are correlated with each other in Figure 15. The surface plot shows a general trend that CS is inversely related to both WA and CIP. All data points in Figure 15, congregate near-surface plot which means CS, CIP and WA are strongly correlated (R2 > 0.8) and models developed to predict these parameters from each other can be formulated for design purposes.

## 4. Conclusions

This study evaluates the influence of a wide range of SF doses on the basic engineering characteristics of high strength concrete. It also explores the modifications of SF-reinforced concrete properties with micro-silica (MS). Following important conclusions can be taken from this research:(1)Fibers have mixed effects on compressive strength (CS). The positive effect of SF on CS is observed only at small doses. A high-volume dose of SF negatively affects CS. MS improves the utilization of SF in advancing the CS of concrete.(2)Similar to CS results, modulus of elasticity (MOE) also shows mixed behavior with varying fiber dose. Low volumes (0.05–0.5%) of SF are beneficial to MOE, whereas high volumes (1–2%) are detrimental to MOE. MS shows synergistic effects with SF on MOE. SF doses of 0.5% produce optimum MOE and CS. The addition of MS is highly useful compared to SF addition if the increment in CS or MOE is desired.(3)Splitting-tensile strength (STS) increases up to 36% with the rising dose of SF (0 to 2%). There is no significant achievement in STS when SF doses increased beyond 1%. STS experience more gain than CS and MOE at all doses of SF. MS improves the net gain in STS due to SF addition. The combined addition of 10%MS and 1%SF produces concrete with 60% more STS than plain concrete.(4)Considering the combined behavior of CS, MOE, and STS, 1%SF can be taken as the optimum dose for high strength concrete.(5)At low to medium fiber doses (0.05–0.25%), the WA of concrete was slightly lower than that of the plain concrete. Whereas high fiber doses (0.5–1%) are determinantal to the imperviousness of concrete. The positive effect of low fiber volumes is very negligible compared to that of the MS. MS can play a key role in downing the WA capacity of high fiber volume concretes.(6)Similar to WA, chloride-ion penetration (CIP) experiences a small reduction of 0.05–0.25% SF. The detrimental effect of high fiber dose can be minimized by MS addition. With the help of 10%MS, 2%SF concrete shows a significant 25% lower CIP compared to plain concrete.

## 5. Future Research

Effect of micro-silica on the interfacial zones of fibers should be studied using scanning electron microscopy. Moreover, combined effect of MS and SF on different strength classes of concrete should also be investigated and compared. Effect of different MS contents on the freeze-thaw and corrosion resistance of SFRC can also be studied.

## Figures and Tables

**Figure 1 materials-13-05739-f001:**
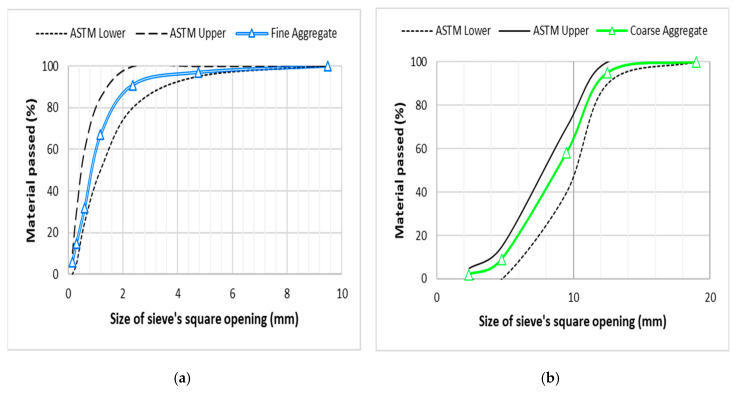
Gradation of (**a**) Siliceous sand/fine aggregate and (**b**) crushed dolomite limestone/coarse aggregate.

**Figure 2 materials-13-05739-f002:**
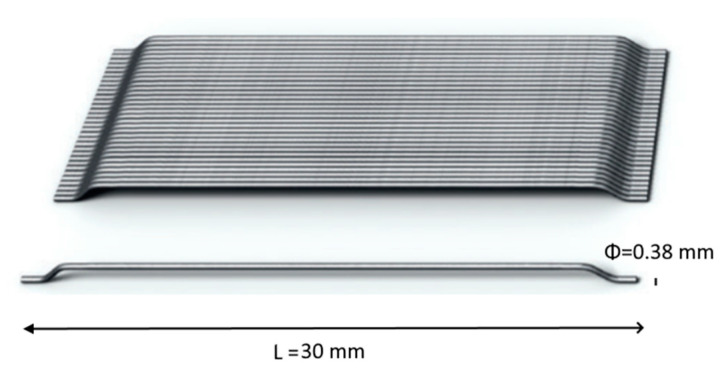
Overview of SF.

**Figure 3 materials-13-05739-f003:**
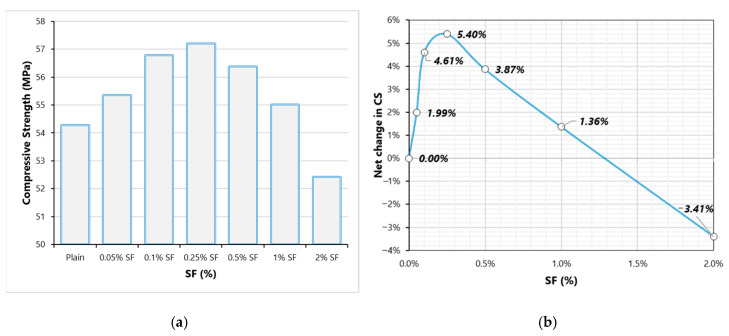
Compressive strength (CS) results (**a**) Variation of CS with SF dosage (**b**) Net change in CS with varying SF dosage.

**Figure 4 materials-13-05739-f004:**
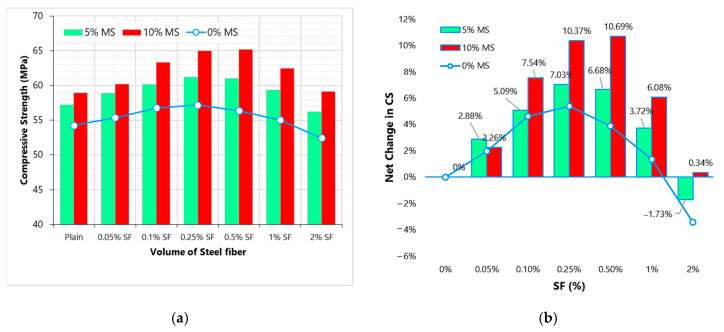
Compressive strength results (**a**) effect of MS on CS with varying SF dosage (**b**) effect of MS on the net change in CS with varying SF dosage.

**Figure 5 materials-13-05739-f005:**
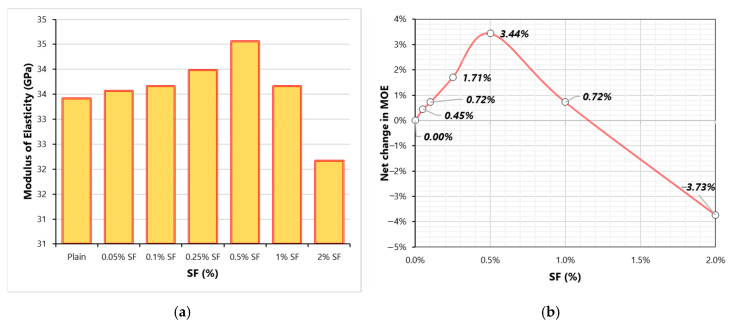
Modulus of elasticity (MOE) results (**a**) Variation of MOE with SF dosage (**b**) Net change in MOE with varying SF dosage.

**Figure 6 materials-13-05739-f006:**
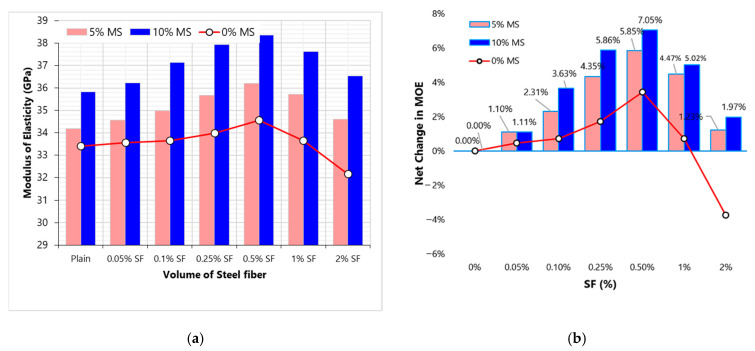
Modulus of elasticity (MOE) results (**a**) effect of MS on MOE with varying SF dosage (**b**) effect of MS on the net change in MOE with varying SF dosage.

**Figure 7 materials-13-05739-f007:**
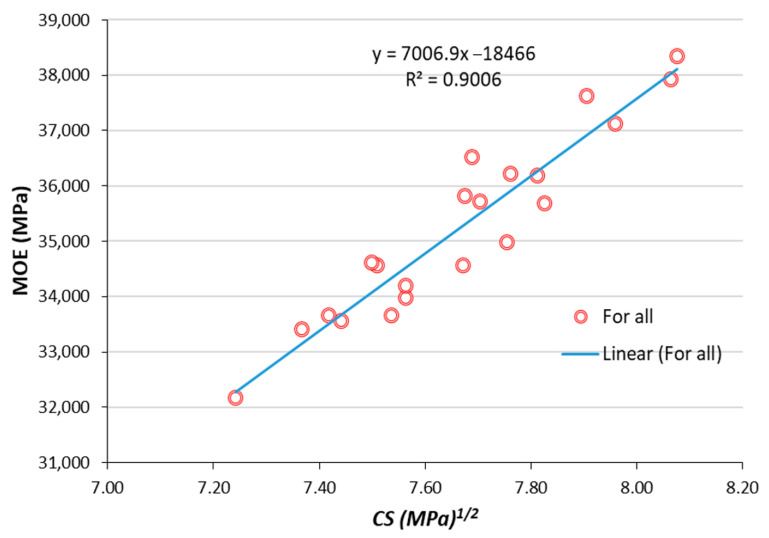
Relationship between MOE and CS^1/2^.

**Figure 8 materials-13-05739-f008:**
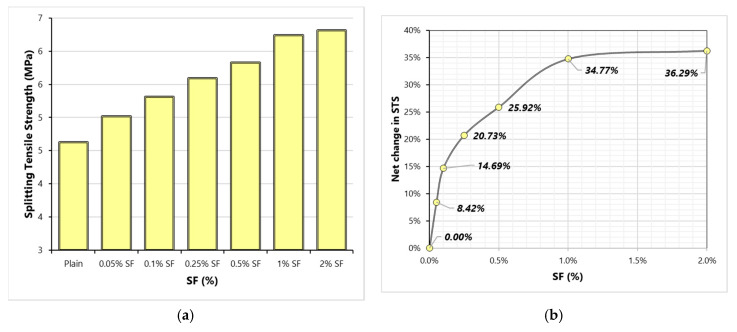
Splitting-tensile strength (STS) results (**a**) Variation of STS with SF dosage (**b**) Net change in STS with varying SF dosage.

**Figure 9 materials-13-05739-f009:**
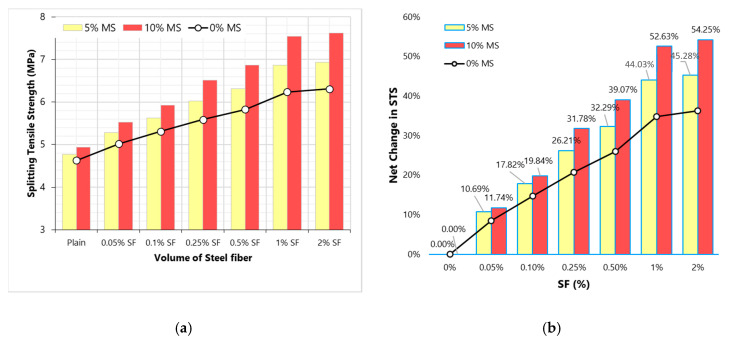
Splitting-tensile strength (STS) results (**a**) effect of MS on STS with varying SF dosage (**b**) effect of MS on the net change in STS with varying SF dosage.

**Figure 10 materials-13-05739-f010:**
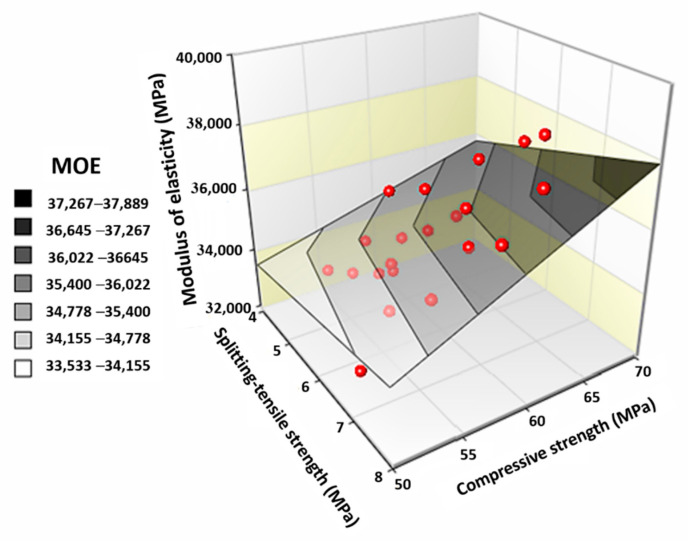
Correlation between mechanical properties (MOE, STS and CS).

**Figure 11 materials-13-05739-f011:**
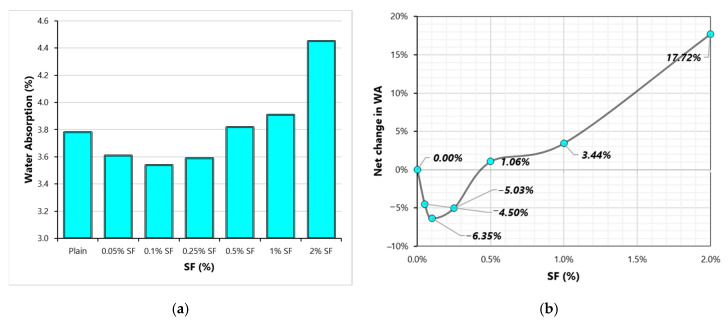
Water-absorption (WA) results (**a**) Variation of WA with SF dosage (**b**) Net change in WA with varying SF dosage.

**Figure 12 materials-13-05739-f012:**
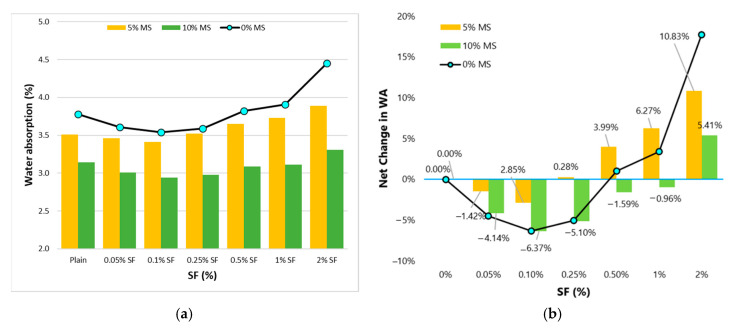
Water-absorption (WA) results (**a**) effect of MS on WA with varying SF dosage (**b**) effect of MS on the net change in WA with varying SF dosage.

**Figure 13 materials-13-05739-f013:**
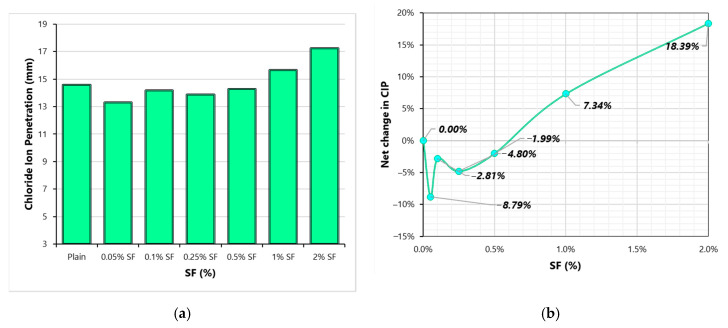
Chloride-ion penetration (CIP) results (**a**) Variation of CIP with SF dosage (**b**) Net change in CIP with varying SF dosage.

**Figure 14 materials-13-05739-f014:**
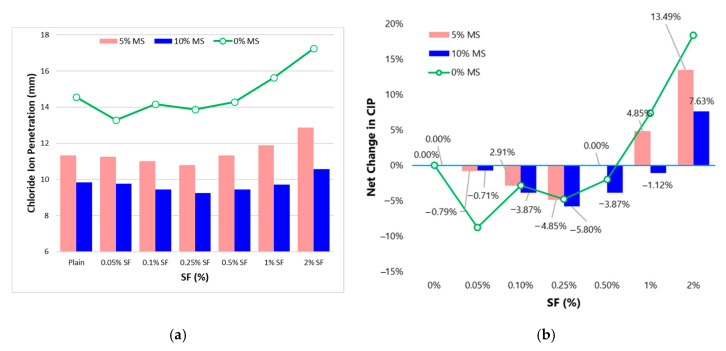
Chloride-ion penetration (CIP) results (**a**) effect of MS on CIP with varying SF dosage (**b**) effect of MS on the net change in CIP with varying SF dosage.

**Figure 15 materials-13-05739-f015:**
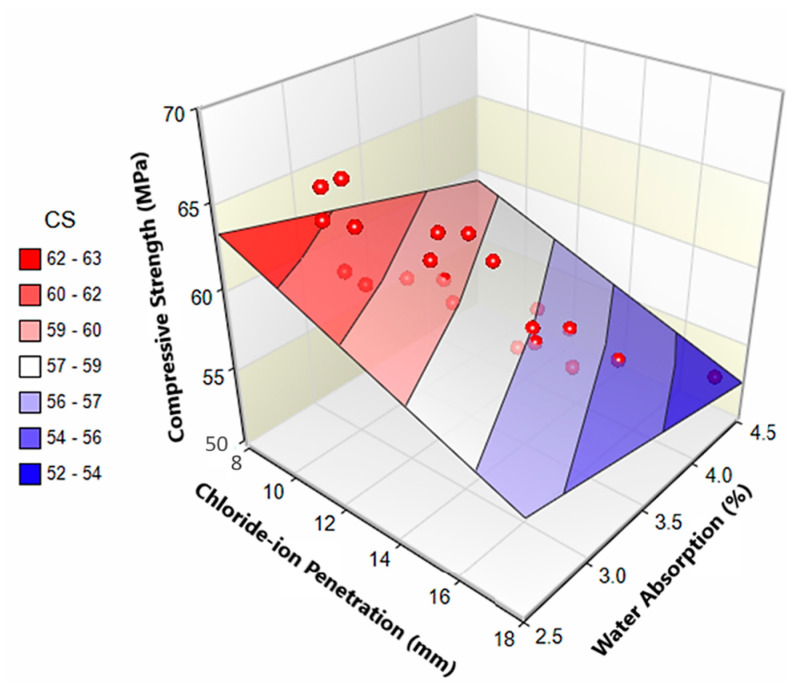
Correlation between CS, CIP and WA.

**Table 1 materials-13-05739-t001:** Characteristics of aggregates.

Aggregate Type	Material	Dry Rodded Density	Water Absorption (%)	10% Fine Value (kN)	Specific Gravity	Maximum Aggregate Size
Fine aggregate	Siliceous sand	1615	0.97	-	2.67	4.75
Coarse Aggregate	Dolomite-sandstone	1519	0.65	148	2.68	12.5

**Table 2 materials-13-05739-t002:** Design of concrete mixes.

Mix No.	Mix ID	MS (%)	SF (%)	Cement (kg/m^3^)	MS (kg/m^3^)	Siliceous Sand (kg/m^3^)	Crushed Limestone (kg/m^3^)	SF (kg/m^3^)	Water (kg/m^3^)	HWR (kg/m^3^)
1	MS0/SF0 (Control)	0	0.00	478	0	657	1077	0	185	2
2	MS0/SF0.05		0.05	478	0	656	1076	4	185	2
3	MS0/SF0.1		0.10	478	0	656	1076	8	185	2
4	MS0/SF0.25		0.25	478	0	654	1074	20	185	2
5	MS0/SF0.5		0.50	478	0	651	1071	39	185	3
6	MS0/SF1		1.00	478	0	644	1064	78	185	3
7	MS0/SF2		2.00	478	0	631	1051	156	185	3
8	MS5/SF0	5	0.00	454	18	657	1077	0	185	2
9	MS5/SF0.05		0.05	454	18	656	1076	4	185	2
10	MS5/SF0.1		0.10	454	18	656	1076	8	185	2
11	MS5/SF0.25		0.25	454	18	654	1074	20	185	2
12	MS5/SF0.5		0.50	454	18	651	1071	39	185	3
13	MS5/SF1		1.00	454	18	644	1064	78	185	3
14	MS5/SF2		2.00	454	18	631	1051	156	185	3
15	MS10/SF0	10	0.00	430	36	657	1077	0	185	2
16	MS10/SF0.05		0.05	430	36	656	1076	4	185	2
17	MS10/SF0.1		0.10	430	36	656	1076	8	185	2
18	MS10/SF0.25		0.25	430	36	654	1074	20	185	2
19	MS10/SF0.5		0.50	430	36	651	1071	39	185	3
20	MS10/SF1		1.00	430	36	644	1064	78	185	3
21	MS10/SF2		2.00	430	36	631	1051	156	185	3

MS: Micro-Silica; SF: Steel Fiber; HWR: High-range Water Reducer.

**Table 3 materials-13-05739-t003:** Overview of mechanical and permeability testing methods and schedule.

Property	Standard Followed	Size of Specimen	Age of Testing
Compressive Strength (MPa)	ASTM C39	100 φ mm × 200 mm cylinder	28 days
Modulus of Elasticity (MPa)	ASTM C469	150 φ mm × 300 mm cylinder	28 days
Splitting Tensile Strength (MPa)	ASTM C496	100 φ mm × 200 mm cylinder	28 days
Water Absorption (%)	ASTM C642	100 φ mm × 50 mm disc	28 days
Chloride Ion Penetration (mm)	Ali et al. [2]	100 φ mm × 100 mm cylinder	28 days curing + 56 days of condition in NaCl solution

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
