# Peer review of "Effect of Varying Steel Fiber Content on Strength and Permeability Characteristics of High Strength Concrete with Micro Silica"

_materials, 2020, doi:10.3390/ma13245739_

Round 1

Reviewer 1 Report

There are some recommendations I made in the text.

Reviewer 2 Report

1) The originality and the scientific value of the subject are good. Indeed, an important problem having direct applications is treated.

2) The Abstract is concrete as it gives the summary of this research work in a concise manner. In addition, it is sufficiently supported by the results obtained during research. 

Nonetheless, given that the overall text contains many abbreviations, it is the reviewer’s opinion that the authors should definitely add a nomenclature at the beginning of the manuscript.

3) The Introduction Section in its current form is not adequate. In this context, I recommend the authors to further analyze and discuss the results of Refs. [1-3], [4-10], [11,12] and [13-15].

Besides, the differences/advantages of the present investigation compared to other literature works should be written out at the end of this Section in a much more detailed and comprehensive manner.

4) The materials, their applications, applied methods and especially the use of the investigated material are explained in detail. The composition, the origin of the material used, dimensions of specimens etc are all mentioned.

5) Presentation of the experimental work is very thorough. Process and prerequisites of sample preparation are clearly mentioned. However, the authors are kindly recommended to provide some further technical details about the laboratory equipment that they used to carry out their experiments.

6) The presentation and clarity of results and data are good. Yet, the discussion of the results is relatively adequate. The authors could give some additional theoretical explanations about Figs. 3 (a,b), 4 (a,b) and 5 (a,b)

In addition, the quality of Fig. 7 is substandard and evidently not in publication level.

The authors are kindly recommended to address this issue.

7) Logic and coherence are concrete and the clarity and quality of writing are sound.

8)  The Conclusions Section performs the findings of this work in a concrete manner.

However, I invite the authors to add a paragraph on the motives and prospects that this work provides for future research.

Overall, it is the reviewer’s opinion that this paper may be recommended for publication provided that the authors interpret these critical remarks in a constructive manner and revise the manuscript accordingly.

Round 2

Reviewer 2 Report

The authors accounted for all critical remarks and made adequate improvements.

Thus I am satisfied with the manuscript in its current form and I recommend it for publication.